# Local tissue mechanics control cardiac pacemaker cell embryonic patterning

Trevor Henley[1,2], Julie Goudy[1,2], Marietta Easterling[1,2], Carrie Donley[3], Robert Wirka[1,2], Michael Bressan[1,2]

**Cardiac pacemaker cells (CPCs) initiate the electric impulses that drive the rhythmic beating of the heart. CPCs reside in a heterogeneous, ECM-rich microenvironment termed the sinoatrial node (SAN). Surprisingly, little is known regarding the biochemical composition or mechanical properties of the SAN, and how the unique structural characteristics present in this region of the heart influence CPC function remains poorly understood. Here, we have identified that SAN development involves the construction of a "soft" macromolecular ECM that specifically encapsulates CPCs. In addition, we demonstrate that subjecting embryonic CPCs to substrate stiffnesses higher than those measured in vivo results in loss of coherent electrical oscillation and dysregulation of the HCN4 and NCX1 ion channels required for CPC automaticity. Collectively, these data indicate that local mechanics play a critical role in maintaining the embryonic CPC function while also quantitatively defining the range of material properties that are optimal for embryonic CPC maturation.**

## Introduction

During each cardiac cycle, a single electrical impulse initiated within the sinoatrial node (SAN) propagates across the heart to coordinate cardiomyocyte contraction. Cardiac pacemaker cells (CPCs) located within the SAN are responsible for rhythmically initiating these electrical impulses. SAN dysfunction is common in humans affecting 1 out of every 600 adults over the age of 65 (Brignole et al, 1990; Mozaffarian et al, 2016) and represents the leading cause for the surgical implantation of mechanical pacemaker devices (Bernstein & Parsonnet, 1996, 2001; Mayosi et al, 1999; Adan & Crown, 2003; Greenspon et al, 2012; Singh et al, 2013; Jensen et al, 2014). Despite the frequency of SAN-related diseases, the underlying cellular pathologies that drive insufficient peacemaking remain largely unknown. Currently, even a basic understanding of how CPCs respond to environmental stimuli known to drive diseases in the rest of the heart is lacking. Consequently, clinical interventionse for the correction of SAN dysfunction are relatively limited. Recent advances in cellular engineering have opened novel strategies for the creation of cellular-based, biological pacing platforms (Miake et al, 2002; Bucchi et al, 2006; Kashiwakura et al, 2006; Boink et al, 2008; Plotnikov et al, 2008; Hashem & Claycomb, 2013; Chauveau et al, 2014; Jung et al, 2014; Cingolani, 2015; Vedantham, 2015; Morikawa et al, 2016; Cingolani et al, 2017; Grijalva et al, 2019; Komosa et al, 2021) which could expand the array of clinical approaches available to correct severe arrhythmic disorders, however, techniques for incorporating engineered CPCs in mature cardiac tissue requires the identification of optimization biomaterials that can support the activity of these cells within an ex vivo, transplantable construct. Thus, developing a quantitative description of the structure–function relationships that exist between CPCs and their environment is of paramount importance to develop the next generation of SAN-related therapeutic interventions.

The SAN is one of the most heterogeneous and structurally diverse subcompartments of the entire heart. First identified by Keith in Flack in 1907 (Keith & Flack, 1907) based on its anatomical characteristics, the SAN has been described as containing small "islands" of nodal cells embedded in a "sea" of connective tissues (Bouman & Jongsma, 1986). Indeed, the atypical microarchitecture present in the SAN has garnered significant interest over the last 100 years (Opthof, 1988; Shiraishi et al, 1992; Csepe et al, 2015, 2016; Ho & Sánchez-Quintana, 2016; Kalyanasundaram et al, 2019). State-of-the-art transcriptional and proteomic profiling have begun to reveal the diversity and specialization of the ECM present in the mature SAN (Liang et al, 2015; Vedantham et al, 2015; Gluck et al, 2017; Li et al, 2019; Linscheid et al, 2019; van Eif et al, 2019; Brennan et al, 2020; Kalyanasundaram et al, 2021; Mandla et al, 2021; Minhas et al, 2021; Okada et al, 2022) which has given rise to the hypothesis that the mechanical properties of the SAN microenvironment may be important for overall CPC function (Gluck et al, 2017; Linscheid et al, 2019; Kalyanasundaram et al, 2021). However, this model has not been empirically tested and few data are available to assess how local biomechanics influence CPC activity.

Here, we have traced the embryological development of the SAN and identified a temporal window during which CPCs acquire a

[1]Department of Cell Biology and Physiology, University of North Carolina at Chapel Hill, Chapel Hill, NC, USA [2]McAllister Heart Institute, University of North Carolina at Chapel Hill, Chapel Hill, NC, USA [3]Department of Chemistry, University of North Carolina at Chapel Hill, Chapel Hill, NC, USA

Correspondence: michael_bressan@med.unc.edu

morphology consistent with the adult SAN. Using next generation sequencing, we further developed a transcriptional profile of the SAN which indicated a dramatic enrichment of encapsulating ECM factors during stages when CPC spatial organization dramatically segregates from the working myocardium. Given the molecular composition of this ECM, we probed the mechanical properties of live cardiac tissue preparations revealing that the developing SAN is significantly softer than the adjacent atrial myocardium. Finally, we fabricated substrates with mechanical properties consistent with various cardiac regions and identified that soft substrates are necessary to sustain CPC functional activity. Collectively, these findings highlight that local biomechanics dramatically influence developmental maturation of the CPC lineage, and our data indicate that approaches designed to construct biological pacemakers for therapeutic uses may need to develop strategies to minimize cellular strain.

# Results

### CPC architecture changes over the course of SAN morphogenesis

Previously, we have reported that SAN morphogenesis is a dynamic process that involves the recruitment of non-muscle mesenchymal cells which invade and surround the pacemaker myocardium (Bressan et al, 2018). To track changes in SAN microarchitecture as this integration process takes place, we used a cell surface marker that we recently identified as labeling CPCs in the chick embryo, FLRT3 (Thomas et al, 2021), and examined CPC cellular arrangement from the end of cardiac looping stages at Hamburger–Hamilton (Hamburger & Hamilton, 1992) (HH) stage 18 (E3 in chick) through the completion of cardiac septation (HH stage 35, E9 in chick). This analysis revealed that at HH 18, CPCs are arranged as a densely packed layer of the myocardium and display a similar morphology to the adjacent atrial chamber myocardium (Fig 1A) (see also [Thomas et al, 2021]). By HH 30, CPCs remodel into a loosely connected network of cells and prominent acellular spaces can be detected within the forming SAN (Fig 1A). This cellular meshwork undergoes further rearrangement between HH 30 and HH 35, as the CPCs reaggregate into small clusters of ~3–10 cells that are separated by collections of non-muscle cells (Fig 1A). Notably, clusters of CPCs seen at HH 35 greatly resemble the morphology of CPCs reported in the adult mammalian heart (Bleeker et al, 1980; Masson-Pevet et al, 1984; Bouman & Jongsma, 1986; Opthof et al, 1986; De Mazière et al, 1992; Sanchez-Quintana et al, 2002), and differ significantly from the cellular architecture present in the adjacent atrial myocardium (Fig 1A). These data suggest that construction of the unique anatomical features present in the mature SAN arise via a series of developmental morphogenetic events that are specific to the forming pacemaker region of the heart.

To examine the biological processes that may control SAN morphogenesis over the above-described developmental window, we isolated HH 30 SAN and atrial samples and performed bulk RNA sequencing. We divided 59 SAN and atrial explants into three technical replicates for this analysis (Fig S1A). To confirm the physiological identity of our samples, we collected six additional

SAN and atrial explants at the time of isolation and performed functional analysis. Importantly, voltage imaging confirmed that SAN explants displayed physiological features consistent with the pacemaker tissue including: uniform slow diastolic depolarization (Fig 1C and D), slow conduction velocity (SAN 5.04 ± 0.47 cm/s, atria 19.4 ± 2.02 cm/s) (Fig 1C and E), rapid cycling rate (SAN 382.48 ± 26.31 ms, atria 682.28 ± 153.15 ms) (Fig 1D and F), and an elongated action potential duration (ADP) when compared with atrial explants (SAN 138.50 ± 3.55 ms, atria 97.88 ± 4.00 ms) (Fig 1D and G). These data demonstrated that the tissue selected for sequencing uniquely displayed physiological characteristics consistent with the SAN.

After functional validation, we confirmed consistency in our RNA sequencing across sample pools by conducting principal component analysis and examining the expression of known positive (*Shox2, Isl1, Tbx3, Hcn4, Flrt3, Lsps1*) and negative (*Nkx2.5, Scn5a, Bmp10, Gja5*) markers of CPCs (Hoogaars et al, 2004, 2007; Blaschke et al, 2007; Espinoza-Lewis et al, 2009; Wiese et al, 2009; Bakker et al, 2012; Ye et al, 2015a, 2015b; Ionta et al, 2015; Liang et al, 2015, 2020; Bressan et al, 2018; Li et al, 2019; van Eif et al, 2019; Thomas et al, 2021; Okada et al, 2022). Marker gene expression analysis indicated that all three sample pools showed similar fold differences in previously identified differentially expressed genes (Fig S1B) and principal component analysis demonstrated strong separation between SAN and atrial sample pools (Fig S1C). In particular, two sample pools displayed high concordance and we focused on these data sets for our downstream bioinformatics analysis (Figs 1H and S1C). Using these data sets, we identified 1,720 genes that were enriched in the SAN when compared with atrial samples and 1,329 genes enriched in atrial samples when compared with the SAN (≥1.5-fold, ≥10 counts per million, $P \leq 0.05$) (Fig 1I and Table S1). To isolate biological processes that were differentially active between the SAN and atria, we performed functional enrichment analysis using g:Profiler (Raudvere et al, 2019) and ShinyGO (Ge et al, 2020) which demonstrated that gene sets associated with ion channel activity, ECM formation, and neuronal/synapse development were up-regulated in the SAN, whereas gene sets related to actin cytoskeletal formation, muscle contraction, and metabolism/mitochondrial activity were generally up-regulated in the atrial samples (Figs 1J and S1D–F).

### SAN morphogenesis results in CPC encapsulation by a proteoglycan-rich ECM

Interestingly, the GO term external encapsulating structure (GO: 0030312) was identified in our above functional enrichment analysis as significantly up-regulated in our SAN RNAseq dataset ($P$-value = $10^{-34}$). Based on the morphological rearrangements that we noted among CPCs between HH 18 and HH 35, we decided to examine the genes associated with this GO term in more detail. We queried all of the differentially expressed genes (either SAN/Atria or Atria/SAN) annotated in the external encapsulating structure gene set (GO:0030312) and focused on transcripts known to encoded ECM proteins. We then categorized these genes based on their known functions, demonstrating that most were associated with collagen biosynthesis, ECM proteoglycans, and/or fibrin clot formation (Fig 2A). We further constructed a protein interaction plot (Snel et al, 2000; Franceschini et al, 2013; Szklarczyk et al, 2015, 2021)

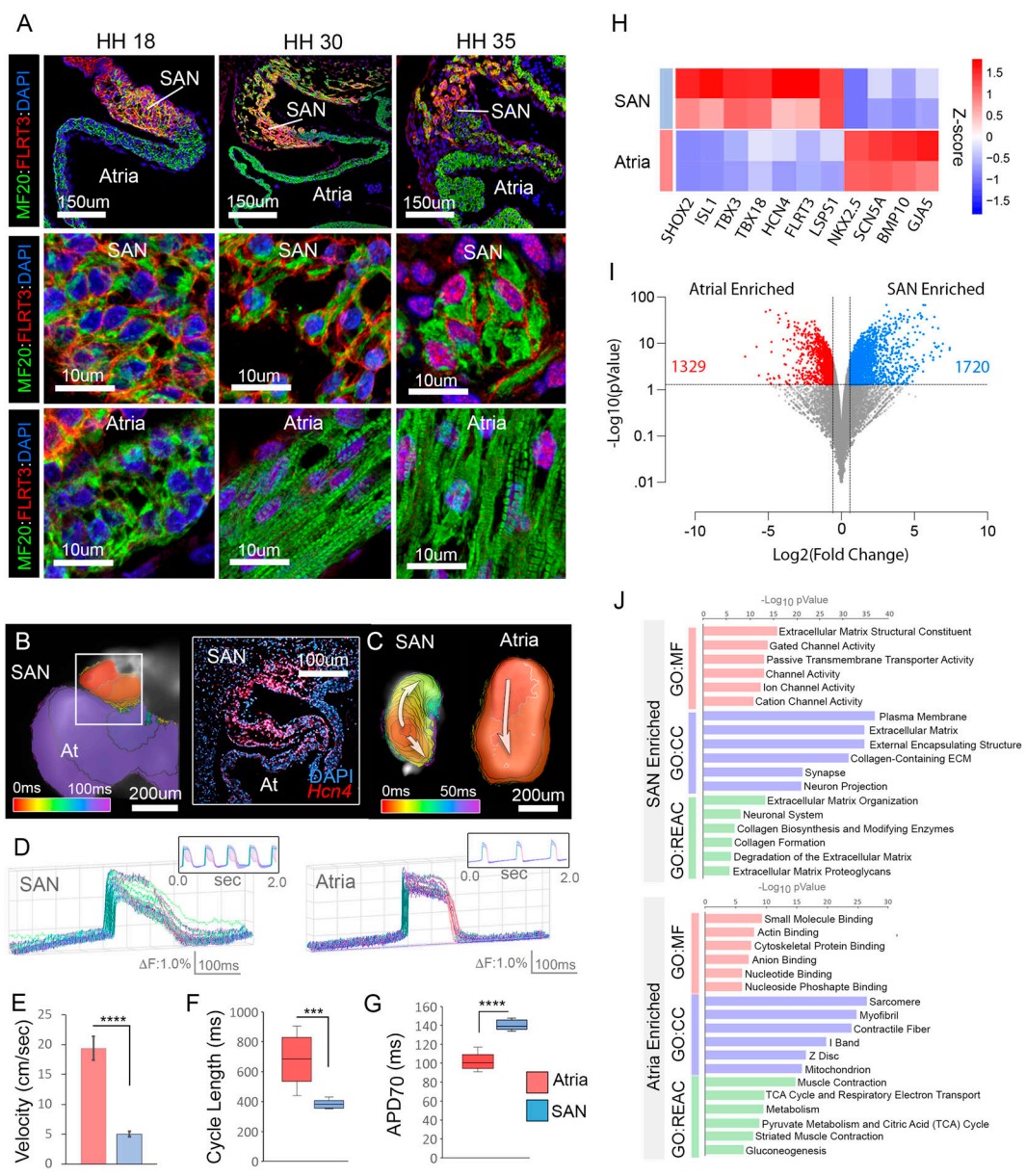

**Figure 1.  Characterization of embryonic SAN morphogenesis.**
**(A)** Staining of the HH18, HH30, and HH35 SAN with the pacemaker cell marker, FLRT3 (red), and muscle marker, MF20 (green). Upper panels: low magnification images with FLRT3-positive cells restricted to the SAN. Middle panels: high magnification images of SAN region. Lower panels: high magnification images of atrial myocardium. **(B)** Voltage imaging of HH30 embryonic atrial (viewed from above). Isochronal map depicts electrical impulse initiation in the SAN (red) and propagation into the atria. The boxed region shows the SAN/atrial junction stained in the cross section with an RNAscope probe against the CPC marker *Hcn4*. **(C)** Isochronal maps of voltage-imaged SAN and atrial explants. **(C, D)** Action potential shapes along the arrows in (C) for SAN and atrial explants. **(E)** Quantification of conduction velocity through SAN and atrial explants (n = 6 per condition). **(F)** Quantification of cycle length in atrial versus SAN explants (n = 6 per condition). **(G)** Quantification of action potential duration (70) in atrial versus SAN explants (n = 6 per condition). **(H)** RNAseq-based analysis of known positive and negative makers of SAN. **(I)** Volcano plot showing differentially expressed genes enriched in the atria (red) and SAN (blue). **(J)** GO term analysis based on SAN and atrial gene expression enrichment. ****$P \leq 0.0001$. Data are represented as mean ± SD.

to define predicted functional interactions between our differentially expressed ECM factors (Fig 2B). Collectively, this analysis identified a core group of ECM proteoglycans enriched in the forming SAN including tenascin C, tenascin R, tenascin N, brevican, and hyaluronan and proteoglycan link protein 1, whereas versican, neuorcan, and hyaluronan and proteoglycan link protein 3 were highly expressed in both tissues (Fig 2B and C). All of these factors

bind to and/or crosslink hyaluronic acid (HA) (Bruckner et al, 1993; Hartig et al, 1994; Miyata et al, 2005; Kwok et al, 2010; Sorg et al, 2016; Chu et al, 2018; Thompson et al, 2018) and our transcriptome analysis further demonstrated that the hyaluronic acid synthetase (Has2) showed ~5.7-fold enrichment in our SAN gene set (Fig 2C). Collectively, these data suggest that a macromolecular ECM with HA as its structural backbone may be enriched in the forming SAN.

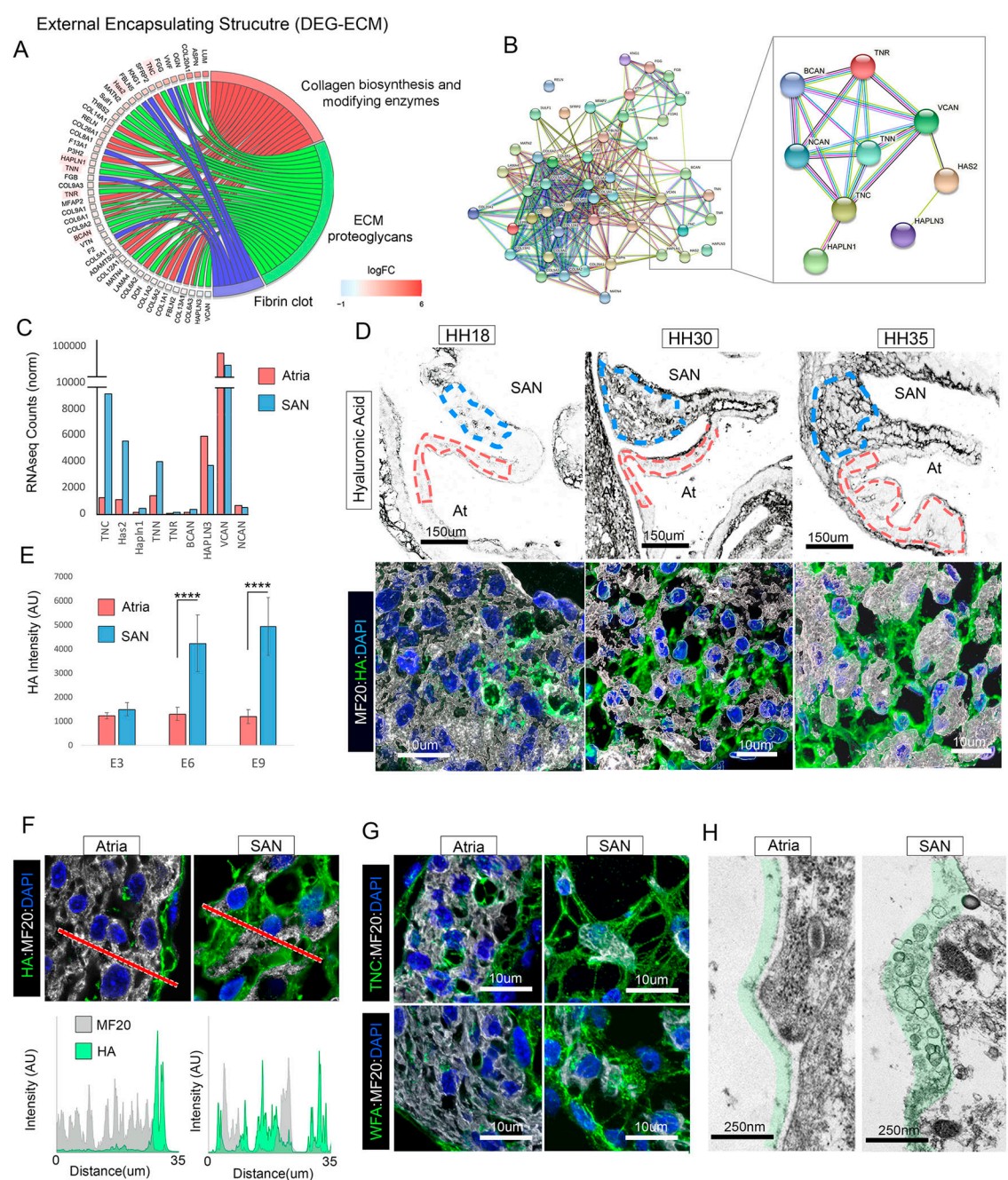

**Figure 2. Matrix deposition during SAN morphogenesis.**
**(A)** Cord plot of differentially expressed genes associated with the GO term external encapsulating structure. **(A, B)** Predicted protein interaction network of the differentially expressed ECM genes from (A). Inset shows a sub compartment of the interaction network focused on HA-binding proteoglycans. **(B, C)** Normalized RNAseq counts of the genes from the inset of (B). **(D)** Upper panels: low magnification of HA deposition in the embryonic SAN/atria at HH18, HH30, and HH35. Bottom panels: high magnification of HA deposition in the SAN. **(D, E)** Quantification of staining intensity for HA across development in atria (red outlined area in (D)) versus SAN (blue outlined area in (D)) (n = 3 hearts per condition). **(F)** Line scan of HA signal intensity in the HH30 atria versus SAN. **(G)** Distribution of TnC and lecticans (stained with WGA) in the HH30 atria versus SAN. **(H)** Transmission electron microscopy images of the surface of HH30 atrial versus SAN cells. Green shading indicates abundant extracellular material on the surface of CPCs. ****$P \leq 0.0001$. Data are represented as mean ± SD.

To validate our RNAseq data, we probed embryonic hearts with a biotinylated HA-binding protein to determine HA distribution. At HH 18, very little HA was detected within the myocardial layer of either the forming SAN or atria (Fig 2D and E). By HH 30, however, HA could be detected in the interstitial spaces surrounding CPCs, but was not present within the myocardial layer of the atria (Fig 2D–F). Notably, in HH 35 hearts, HA-positive material was detected within the myocardial layer of the SAN where it localized in between and along the surface of large CPC clusters. However, very little HA was present in the myocardial layer of the atria, instead being largely restricted

to the epicardial/endocardial surfaces of the tissue (Fig 2D and E). To compare HA distribution to collagen matrix formation, we stained SAN/atrial tissue for Col3A1. Unlike HA, which was abundantly deposited in SAN interstitium by HH30, Col3A1 was mainly detected along the endocardial surface of the SAN. At HH 35 Col3A1 became enriched along the epicardial surface of both the SAN and atrial and was abundantly deposited in the terminal sulcus separating CPCs from the adjacent atria (Fig S2A). These data demonstrate that the HA-based ECM in the forming SAN did not directly overlap with Col3A1 localization.

We further explored the localization of HA-interacting factors including tenascin C and lecticans (eg. neurocan, brevican, and versican) which are labeled by *Wisteria floribunda* agglutin (Hartig et al, 1994). In agreement with our HA staining above, tenascin C was detected in the SAN interstitium at both HH 30 and HH 35 (Figs 2G and S2B) and *Wisteria floribunda* agglutin-positive material was detected in association with non-muscle cells ensheathing CPCs (Figs 2G and S2B). Furthermore, ultrastructural analysis demonstrates that CPCs in the HH 30 SAN were surrounded with a dense pericellular matrix that was much less prominent within the atrial myocardium (Fig 2H).

## CPC encapsulation results in a mechanically compliant embryonic SAN

HA-based matrices are known to have low shear stiffness and high viscoelasticity (Kreger & Voytik-Harbin, 2009; Cowman et al, 2015). Therefore, we hypothesized that the forming SAN may have different biomechanical properties than the adjacent atrial tissue. Initially, we tested this on a macroscopic scale by mounting explants of E6 SAN or right atrial free wall under a micropipette attached to a digitally controlled micromanipulator. We live imaged the explants during a programed 150 $\mu$m indentation and tracked tissue deformation (Fig 3A and Video 1). This revealed that much larger deformation fields were detected in the SAN relative to the atria suggesting softer SAN tissue properties. To determine the actual elastic modulus of the tissue, we probed SAN and atrial explants using a nanoindenter. Each sample was probed over an 80 × 80 $\mu$m region and Young's Modulus was calculated by fitting load/indentation curves based on a Hertzian model (Lieber et al, 2004; Oyen & Cook, 2009; Qian & Zhao, 2018; Kontomaris & Malamou, 2020). This revealed that the SAN was ~5–10-fold softer than the adjacent atria (SAN = 461.2 ± 364.1 Pa versus atria = 2,790.0 ± 2,010.6 Pa) (Fig 3B–E). Three-dimensional reconstruction of the regions probed for nanoindention confirmed that CPCs in the SAN were surrounded by dense HA-positive material (Fig 3F and G).

Given the significant differences between the mechanical properties of the forming SAN and atria, we next examined how substrate stiffness influenced CPC morphology. We fabricated fibronectin-coated polyacrylamide (PA) gels with stiffnesses corresponding to the embryonic SAN (~200 Pa) and atria (~2,700 Pa) (Fig 3H) (Tse & Engler, 2010; Chopra et al, 2011; Thomas et al, 2021). In addition, we also generated PA gels with stiffnesses matching the healthy adult ventricular myocardium (~22,000 Pa) and diseased adult ventricular myocardium (~50,000 Pa) (Berry et al, 2006; Bhana et al, 2010; Majkut et al, 2013; Chiou et al, 2016) as the ventricles are often targeted as the implantation site for tissue-engineered

pacemaker cells (Cingolani et al, 2017; Komosa et al, 2021). Notably, CPCs plated on 200 Pa gels retained the rounded morphology consistent with CPCs in vivo (Fig 3I–L). In contrast, CPCs plated on PA gels of 2,700 Pa or above became elongated and underwent significant hypertrophy (increase surface area and volume) when compared with CPCs in vivo (Fig 3I–L).

## Increasing environmental stiffness results in CPC dysfunction

To test if CPC function was influenced by local mechanics, we loaded CPCs plated on PA gels (200–50,000 Pa) with the fluorescent calcium indicator Cal520 and monitored calcium transient oscillation. Importantly, ~80% of CPCs plated on 200 Pa gels displayed rhythmic calcium transients (Fig 4A–C). In contrast, ~60% of CPCs on stiffer gels displayed disorganized calcium sparks that did not cohere into whole cell transients (Video 2). Furthermore, the low proportion of active cells on 2,700 Pa and above (~40%) displayed calcium waves that initiated and propagated slowly across the cell (Fig 4A and B). To compare functional characteristics among our isolated CPCs with activity within the SAN, we transfected constructs encoding a membrane-localized RFP and the genetically encoded calcium indicator GCaMP6F into the heart (Fig S2C and D). This allowed us to mosaically track calcium transient behavior in individual CPCs in vivo (Goudy et al, 2019) and compare behavior with cells cultured on various hydrogels (Fig 4D). We then quantified calcium transient duration and cycle length across conditions. CPCs plated on 200 Pa gels displayed an average calcium transient duration of 306.2 ± 85.9 ms (compared with 265.6 ± 38.2 ms in vivo), and a cycle length of 666.1 ± 193.2 ms (compared with 526.8 ms ± 58.9 in vivo) (Fig 4E–H). In contrast, calcium transient durations and cycle lengths were significantly elongated in cells cultured 2,700 Pa gels and above (Fig 4E–H). These data demonstrate that CPC function is sensitive to local mechanics and that increasing substrate stiffness results in disrupted CPC oscillatory behavior.

## HCN4 and NCX1 channel distribution is disrupted on stiffer substrates

To determine possible mechanisms underlying the breakdown of CPC function on stiffer materials, we examined the expression of two channels that are critical for CPC automaticity, HCN4, and NCX1 (gene name *Slc8a1*) (Stieber et al, 2003; Ueda et al, 2004; Zicha et al, 2005; Nof et al, 2007; Baruscotti et al, 2011, 2017; Gao et al, 2013; Groenke et al, 2013; Herrmann et al, 2013; Chen et al, 2014; Verkerk & Wilders, 2015; Kozasa et al, 2018; Bychkov et al, 2020; Yue et al, 2020). CPCs were plated on PA gels ranging from 200–22,000 Pa and RNAscope-based fluorescent in situ hybridization was used to quantify *Hcn4* and *Ncx1* transcript levels at single-cell resolution. This analysis revealed that *Hcn4* transcript levels drop from ~52.0 ± 18.1 puncta per cell on 200 PA substrates to 17.4 ± 14.2 puncta per cell when substrate stiffness increases to 2,700 Pa or above (Fig 5A and B). In contrast, no differences in the absolute levels of *Ncx1* transcript were noted across our conditions (Fig 5A and C). We also examined *Hcn4* levels in atrial myocytes plated on different substrate stiffnesses. Embryonic atrial cells from HH 30 do not natively express high levels of *Hcn4* (See Fig 1B), however, when plated on 200 Pa substrates, we saw a significant induction of *Hcn4* transcript

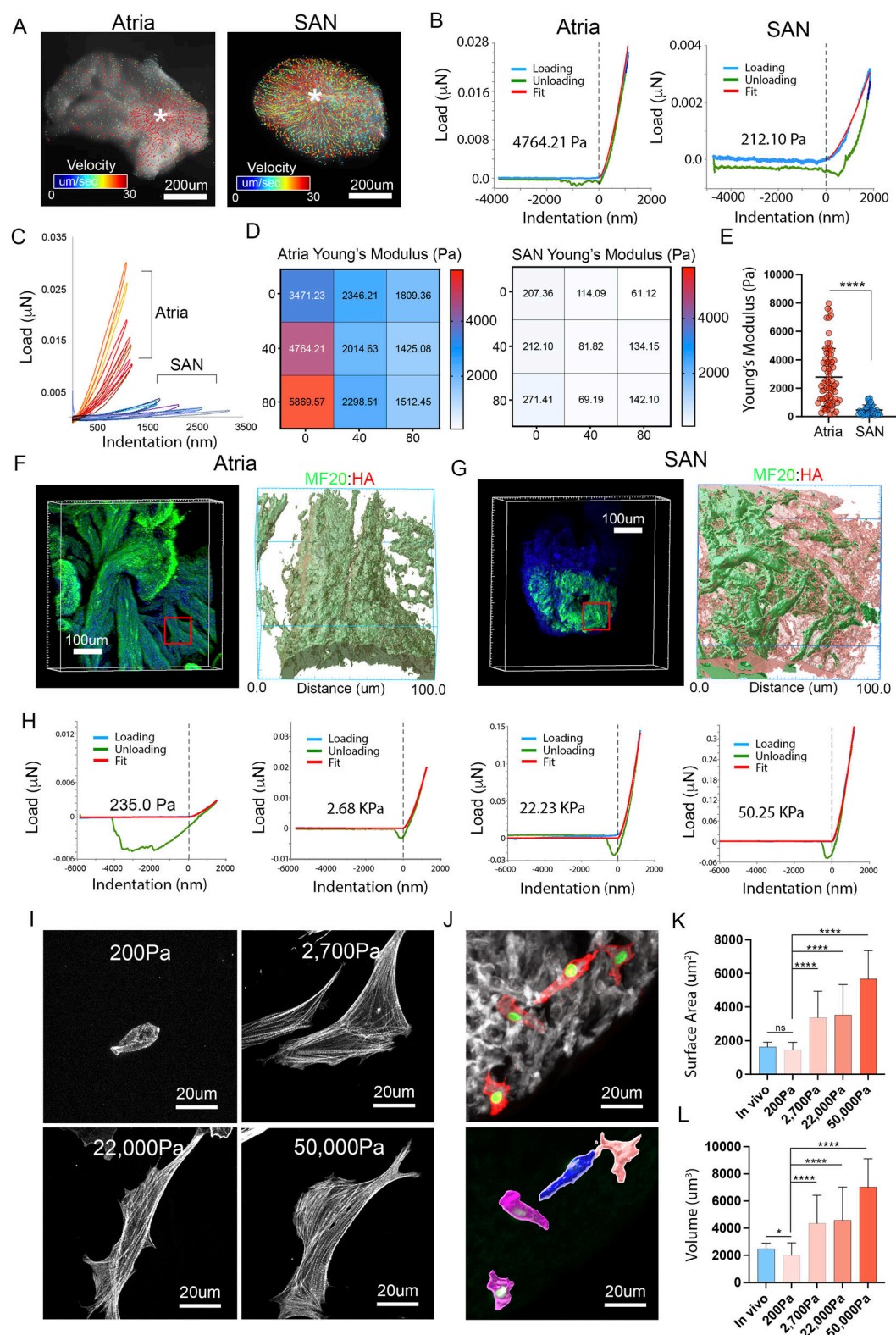

**Figure 3. Cellular mechanics of the embryonic SAN.**
**(A)** Velocity plots charting the movement of the tissue surface after a 150 μm indentation (see also Video 1). **(B)** Representative load/indentation plots for the atria free wall and SAN. Curves are fit with a Hertzian model (red) to calculate Young's Modulus. **(C)** Load indentation plots for nine indentations performed on the atria and SAN of the same heart. **(C, D)** Relative position and calculated Young's Modulus of the data from (C). **(E)** Quantification of calculated Young's Modulus from atria versus SAN (n = 107 measurements from eight hearts). **(C, D, F)** Volumetric reconstruction showing MF20 (green) and HA (red) in the atrial muscle probed by nanoindentation in (C, D). Inset

when compared with 2,700 Pa or 22,000 Pa substrates (Fig S3C and D). Collectively these data demonstrate that *Hcn4* transcript levels are sensitive to local mechanics.

We next examined protein levels for both channels. For these studies, we generated a monoclonal antibody against chick HCN4 (Fig S4). In agreement with the drop in *Hcn4* transcript levels, we detected lower staining intensity for HCN4 protein in cultured CPCs plated on either 2,700 Pa or 22,000 Pa substrates when compared with those cultured on 200 Pa gels (Fig 5D and F). Interestingly, NCX1 staining intensity also dropped in CPCs cultured on stiffer substrates despite our data indicating that transcript levels were relatively preserved (Fig 5D and H). Beyond overall intensity, we also noted that the localization of both HCN4 and NCX1 changed dramatically on stiffer substrates. Although CPCs cultured on 200 Pa gels displayed broad HCN4/NCX1 distribution across the cell surface, both proteins became restricted to subdomains of the membrane on 2,700 and 22,000 Pa gels (Fig 5E). To quantify this, we normalized HCN4 and NCX1 staining area to total cell volume (determined by MF20 staining). This revealed that the area of HCN4 and NCX1 staining dropped significantly between 200 Pa and 2,700 and 22,000 PA substrates with both proteins colocalizing to discreet overlapping regions of the cell surface (Fig 5E, G, and I). To determine if the redistribution of ion channel localization was a generalized response to altered substrate stiffness, we examined Nav1.5 in atrial myocytes. Nav1.5 is the main voltage-gated sodium channel found in the working myocardium and is highly expressed by atrial myocytes (Tellez et al, 2006; Abriel, 2007; Tfelt-Hansen et al, 2010). In contrast to Hcn4 and NCX1 in CPCs, we saw no change in the distribution of Nav1.5 as substrate stiffness was increased from 200–50,000 Pa in atrial myocytes (Fig S3E). Collectively these data demonstrate that the trafficking and/or localization of critical ion handling proteins required for CPC automaticity specifically become dysregulated under conditions where local environmental stiffnesses exceed those present in the SAN.

# Discussion

The SAN is a structurally distinct and complex region of the heart. A consistent SAN characteristic noted across a broad spectrum of vertebrates is the presence of small clusters of CPCs interwoven into a heterogeneous ECM meshwork (Keith & Flack, 1907; Lev, 1954; James, 1961, 1977; Van Mierop & Gessner, 1970; Bleeker et al, 1980; Woods et al, 1982; Masson-Pevet et al, 1984; Bouman & Jongsma, 1986; Opthof et al, 1986; De Mazière et al, 1992; Kohl et al, 1994; Boyett et al, 2000; Zhang et al, 2001; Camelliti, 2004; Matsuyama et al, 2004; Shimada et al, 2004; Chandler et al, 2009; Monfredi et al, 2010; Nikolaidou et al, 2012; Csepe et al, 2015; Wen & Li, 2015; Csepe et al,

2016; Lang & Glukhov, 2018; Kalyanasundaram et al, 2019; Lang & Glukhov, 2021; Okada et al, 2022). In this report, we sought to investigate the developmental construction of the SAN microenvironment, characterize its biomechanical properties, and determine whether local tissue mechanics influence embryonic CPC function. Herein, we identified a developmental window during which mature CPC cellular architecture emerges and uncovered that this process coincides with the enrichment of genes associated with extracellular encapsulation. Furthermore, we identified that a HA-based, proteoglycan-rich ECM surrounds CPCs during SAN morphogenesis and this region of the heart takes on mechanical properties far softer than the adjacent atrial WM. Finally, we demonstrated that subjecting embryonic CPCs to substrate stiffnesses higher than the forming SAN results in loss of coherent electrochemical oscillation and disruption of ion channel localization. These data have uncovered that the unique microenvironment present in the SAN is built during late cardiac development and that embryonic CPC functional optimization is directly influenced by local mechanics.

Our sequencing data identified a collection of interacting ECM factors are specifically expressed in the forming SAN including hyaluronic acid synthetases, hyaluronic acid link proteins, tenascins, and lecticans. These factors make up the major constituents of the brain ECM and assembly into highly charged perineuronal nets that surround fast spiking excitatory neurons and can control ion channel localization, influence membrane capacitance, and can modulate ion diffusion in the extracellular space (Bruckner et al, 1993; Hartig et al, 1994; Miyata et al, 2005; Sorg et al, 2016; Chu et al, 2018). In the current study, we did not assess the influence the surface chemistry of this matrix may have on CPC function, though this is a necessary line of investigation for future studies. We did, however, confirm that the embryonic SAN is far more mechanically compliant than the adjacent atria consistent with the predicted biomechanical properties of an HA-based matrices and the soft ECM present in the brain (Axpe et al, 2020; Budday et al, 2020).

Proteomic and next-generation sequencing studies conducted in adult mouse and human SAN samples have indicated the presence of factors associated with a highly elastic ECM (Gluck et al, 2017; Linscheid et al, 2019; Kalyanasundaram et al, 2021). This has raised the possibility that that CPCs reside in a microenvironmental niche that mechanically insulates them from the hemodynamic and contractile forces experienced by working myocardium. Our studies in the embryonic heart are consistent with this model, particularly given our measured Young's modulus being between 5 and 10 times lower in the SAN than in the atria. However, our data are not in perfect agreement with measurements taken from the adult porcine SAN which, to our knowledge, is the only other comparable measurement of SAN mechanical properties (Gluck et al, 2017). In this previous report, the authors decellularized the SAN and measured the stiffness of the remaining ECM scaffold using atomic

---

shows high magnification reconstruction of the area probed. **(C, D, F, G)** As in (F), for the SAN measurements from (C, D). **(H)** Nanoindentation of polyacrylamide gels generated to mimic various cardiac tissue stiffnesses (see text). **(I)** Morphology of CPCs plated on PA gels of 200, 2,700, 22,000, and 50,000 Pa stiffnesses. **(J)** Morphology of individual CPCs labeled with a membrane-targeted RFP and nuclear-targeted eGFP within the SAN of a HH30 heart. **(K)** Quantification of the CPC surface area in vivo versus various stiffness gels (n = 103 cells). **(L)** Quantification of CPC volume in vivo versus various stiffness gels (n = 103 cells). ****$P ≤ 0.0001$, *$P ≤ 0.05$. Data are represented as mean ± SD.

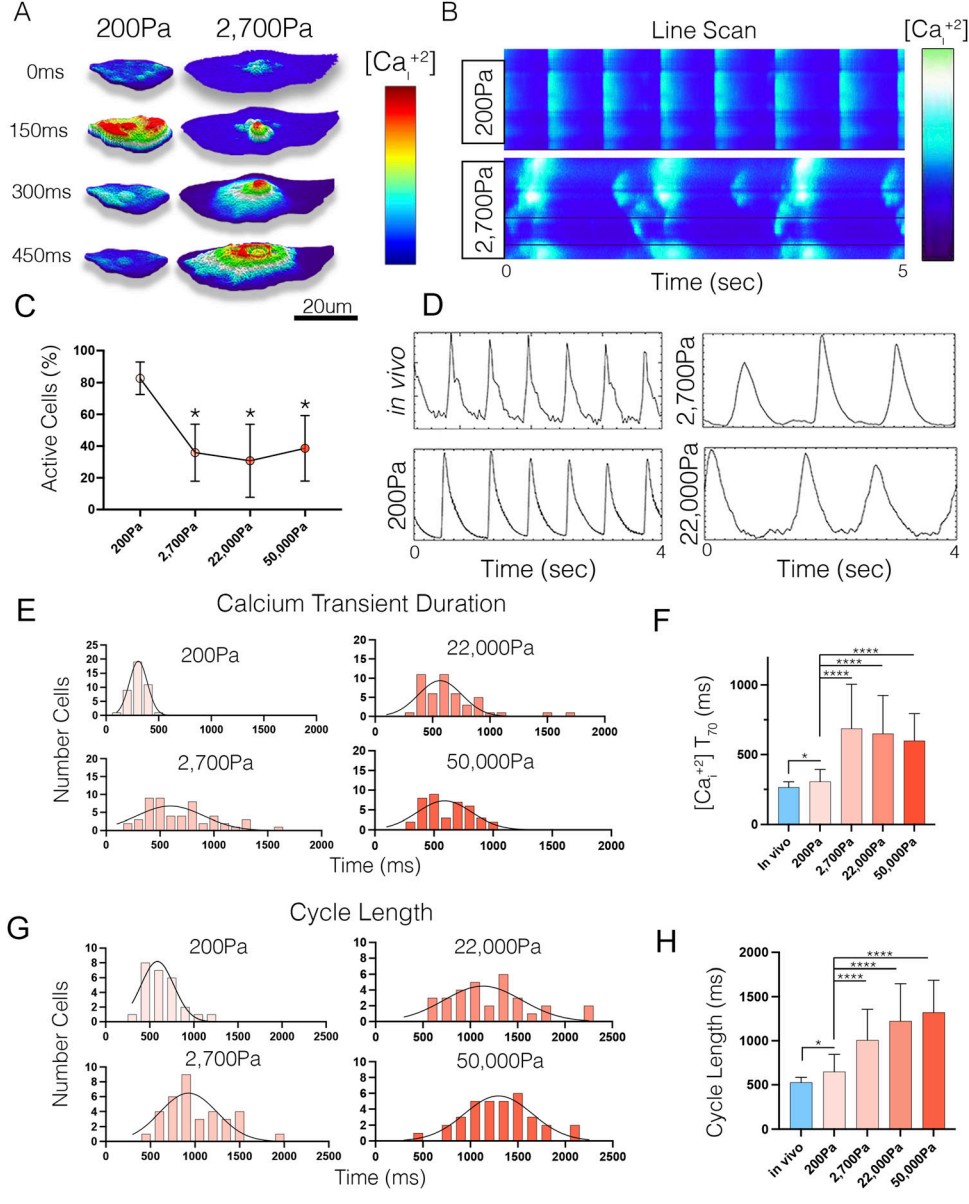

**Figure 4. Breakdown of CPC activity on Stiff substrates.**
**(A)** Time series of calcium transient activity in CPCs plated on 200 versus 2,700 Pa gels. Note rapid increase and clearance of calcium in CPC on 200 Pa gel versus slow propagation of intracellular calcium wave in CPC on 2,700 Pa gel. **(B)** Line scan of CPC on 200 versus 2,700 Pa gel showing loss of coordinated intracellular calcium handling. **(C)** Quantification of the percentage of active cells (showing whole cell calcium transients) on various stiffness substrates (n = 948 cells). **(D)** Comparison of calcium transients between in vivo, 200, 2,700, and 22,000 Pa stiffnesses (n = 128 cells). **(E)** Quantification of calcium transient duration (70) in vivo versus on 200, 2,700, 22,000, and 50,000 Pa substrates. **(E, F)** Distribution of calcium transient durations form CPCs from (E). **(E, G)** as in (E) for calcium transient peak-to-peak cycle length. **(G, H)** Distribution of the recorded cycle lengths from (G). ****$P \leq 0.0001$, *$P \leq 0.05$. Data are represented as mean ± SD.

force microscopy. Their data indicated that the collagen-rich SAN ECM that remained after decellularization was stiffer than that of the left ventricle. The authors concluded that high tensile strength of the collagen scaffold surrounding CPCs would provide a rigid frame protecting CPCs from mechanical strain. Furthermore, as CPCs were proposed to have low integrin content, they would not be expected to interact with the rigid frame in which they were encased (Gluck et al, 2017). Thus, despite differences in data, the overriding conclusion of this prior study is consistent with a model in which the SAN microenvironment mechanically isolates CPCs. Our data may differ from this previous report for a number of reasons including stages (embryonic versus adult), technique employed (atomic force microscopy versus nanoindentation) or condition (decellularized tissue versus live preparations). Although we cannot currently reconcile the differences between our data

and those obtained from the decellularized adult porcine SAN, it is reasonable to speculate that the SAN may form as a very soft tissue which is optimal for CPC maturation (while the cells are still fairly plastic) and that a rigid scaffold may then encase this structure to further mechanically isolate the cells.

Finally, our data demonstrate that when cultured on substrates with stiffnesses corresponding to the embryonic atria, the adult ventricle or an infarcted boarder zone, embryonic CPCs display compromised electrochemical oscillation. On stiffer substrates, ~60% of CPCs displayed discontinuous calcium sparks/wavelets that did not coalesce into a whole cell transient. Furthermore, the ~40% of CPCs that did show relatively normal activity on stiffer substrates displayed disrupted calcium propagation across the cell and/or elongated calcium transient durations. This disrupted function is consistent with the restricted localization of both HCN4

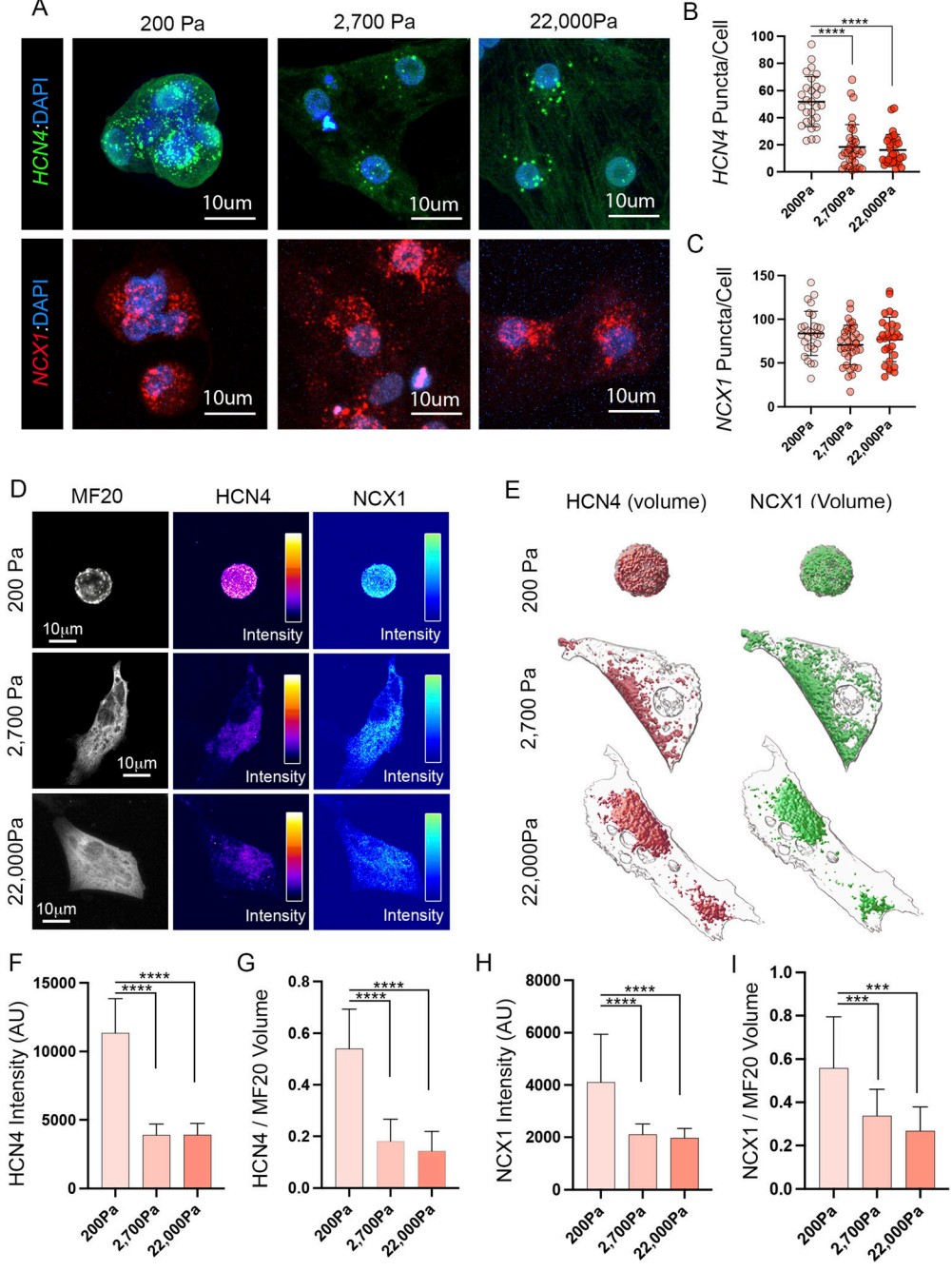

**Figure 5. Loss of HCN4 and NCX1 on stiffer substrates.**
**(A)** RNAscope staining using probes against *Hcn4* (green) and *NCX1* (red) in CPCs plated for 48 h on 200, 2,700, and 22,000 PA substrates. **(B)** Quantification of *Hcn4* RNAscope signal (puncta per cell, n = 93 cells). **(C)** Quantification of *NCX1* RNAscope signal (puncta per cell, n = 93 cells). **(D)** Antibody staining intensity for HCN4 and NCX1 (pseudo colored) for CPCs plated for 24 h on 200, 2,700, and 22,000 Pa substrates. **(E)** Volumetric reconstructs of HCN4 and NCX1 protein distribution in CPCs plated on 200, 2,700, and 22,000 Pa substrates for 24 h. **(F)** Quantification of HCN4 staining intensity (n = 60 cells). **(G)** Quantification of HCN4 positive volume per cell normalized to MF20 (n = 61 cells). **(H)** Quantification of NCX1 staining intensity (n = 66 cells). **(I)** Quantification of NCX1 positive volume per cell normalized to MF20 (n = 54 cells). ****$P \leq 0.0001$, ***$P \leq 0.001$. Data are represented as mean ± SD.

and NCX1 that we observed on stiffer substrates. HCN4 is a core component of the cell surface voltage-gated ion channel system required for CPC automaticity and NCX1 links intracellular calcium oscillations with slow diastolic depolarization (Lakatta et al, 2010; Tsutsui et al, 2018; Thomas et al, 2021). Loss of either of these channels severely disrupts the SAN function (Baruscotti et al, 2011; Groenke et al, 2013). Our data demonstrate that on stiffer substrates, both channels become localized to subfractions of the membrane surface, indicating that organization of the multi-channel cycling units required for embryonic CPC

function is sensitive to local biomechanics. Although future studies will be required to identify the mechanosensitive trafficking programs that control HCN4/NCX1 localization, the current study demonstrates that both channels display mechanosensitive behaviors in maturing CPCs. Furthermore, our data provide the first direct insight into the range of microenvironmental stiffnesses that can be tolerated by developing CPCs. Importantly, these values indicate that future strategies for the development of cellular-base biological pacemakers for therapeutic use may require techniques for

lowering local mechanical strain to ensure the sustained long-term function of engineered pacemaker-like cells.

# Materials and Methods

### Animals

Fertilized chicken eggs were obtained from Allen Harim Hatchery and placed in a humidified incubator (Hova-Bator, Genesis 1588) until desired developmental Hamburger–Hamilton Stages as follows: HH stage 18 for E3, HH stage 30, for E6, and HH stage 35 for E9 (Hamburger & Hamilton, 1992). All procedures were approved by the University of North Carolina's American Association for Accreditation of Laboratory of Animal Care Committee.

### Immunohistochemistry

Cardiac tissue was isolated and fixed in 4% PFA for 2 h at 4°C. The tissue was then washed 3 × 5mins in 1X DPBS (14190144; Gibco) and then dehydrated though a graded ethanol series, transitioned into xylene, and then embedded in paraffin. 12 $\mu$m thick sections were then cut and stored at room temperature until use. For immunohistochemistry, sections were deparaffinized and rehydrated following standard procedures and antigen retrieval was performed using sodium citrate (Thomas et al, 2021). Slides were blocked in 1X DPBS (14190144; Gibco) with 1% BSA (BP9703100; Fisher BioReagents) and 0.1% Tween-20 (85113; Thermo Fisher Scientific) for 1 h at room temperature and then incubated with primary antibodies overnight at 4°C. After three washes in 1X DPBS, secondary antibodies were added for 1 h at RT followed by three washes in 1X DPBS. The slides were mounted with aqueous mounting media containing DAPI (ab104139; Abcam).

Cell cultures were fixed with 4% PFA for 30 min at RT, followed by 3 × 10-min washes in 1X DPBS (14190144; Gibco). Cells were blocked in 1X DPBS (14190144; Gibco) with 1% BSA (BP9703100; Fisher BioReagents) and 0.1% Tween-20 (85113; Thermo Fisher Scientific) for 1 h at RT followed by primary antibodies overnight at 4°C. After three washes in 1X DPBS, secondary antibodies added for 1 h at room temperature followed by three washes in 1X DPBS.

Cryosections were used for HCN4 antibody validation. As above, hearts were isolated and fixed in 4% PFA. The tissue was then washed in 1X DPBS (14190144; Gibco), propagated through a graded sucrose series (10%, 30%, 50%), and then embedded in OCT. 12 $\mu$m sections were then cut on a cytostat and stored at –80°C until use. For staining, sections were washed three times in 1X DPBS (14190144; Gibco) then blocked in in 1X DPBS (14190144; Gibco) with 1% BSA (BP9703100; Fisher BioReagents) and 0.1% Tween-20 (85113; Thermo Fisher Scientific) for 1 h at RT followed by primary antibody (HCN4 2B2) overnight at 4°C. After three washes in 1X DPBS, secondary antibodies were added for 1 h at RT followed by three washes in 1X DPBS. The slides were mounted with aqueous mounting media containing DAPI (ab104139; Abcam).

Whole tissue explants were fixed with 4% PFA for 2 h at 4°C, followed by 5 × 10-min washes in 1X DPBS (14190144; Gibco). Samples were blocked in 1X DPBS (14190144; Gibco) with 1% BSA (BP9703100; Fisher BioReagents) and 0.1% Tween-20 (85113; Thermo Fisher Scientific) for 1 h at RT followed by primary antibodies overnight at 4°C. The samples were then rinsed three times with 1X DPBS (14190144; Gibco) and washed six times for 30 min at RT. Secondary antibodies and DAPI were added overnight at 4°C followed by three rinses in 1X DPBS (14190144; Gibco) and several 30-min washes (dark) at RT until imaging.

### Antibodies and staining reagents

The following reagents were used for immunohistochemistry: MF20 (14650380; Invitrogen) (1:500); FLRT3 (PA563240; Invitrogen) (1:250); hyaluronic acid binding protein, bovine nasal cartilage, biotinylated (38591150UG; MilliporeSigma) (1:100); Wisteria floribunda lectin biotinylated (B13552; Vector Laboratories) (1:100); Col3A1 (3B2-s; DSHB) (1:250), TNC (LS-C39574-50; LSBio) (1:500); Nav1.5 (LS-C30483; LSBio) (1:500), phalloidin-647 (A22287; Invitrogen) (1:40); NCX1 (LS-C73201-100; LSBio) (1:500); streptavidin-568 (S11226; Invitrogen) (1:500); DAPI (62248; Thermo Fisher Scientific) (1:1,000); IgM (heavy chain) cross-adsorbed goat anti-mouse, Alexa Fluor 555 (A21426; Invitrogen) (1:500); IgG1 cross-adsorbed goat anti-mouse, Alexa Fluor 647 (A21240; Invitrogen) (1:500); IgG2b cross-adsorbed goat anti-mouse, Alexa Fluor 488 (A21141; Invitrogen) (1:500); IgG (H+L) cross-adsorbed goat anti-rabbit, Alexa Fluor 568 (A11011; Invitrogen) (1:500); goat anti-mouse IgG2b cross-adsorbed secondary antibody, Alexa Fluor 647 (A21242; Invitrogen) (1:500); goat anti-rabbit IgG (H+L) cross-adsorbed secondary antibody, Alexa Fluor 488 (A11008; Invitrogen) (1:500); goat anti-mouse IgM mu chain Alexa Fluor 488 (ab150121; Abcam) (1:500).

The anti-chick HCN4 antibody was developed for this study. A custom mouse monoclonal antibody was made against amino acids 1,087–1,100 of the chick HCN4 protein (XP_040536079) by Precision Antibody (Fig S3A). After initial screening using ELISA, several monoclonal clones were tested for specific labeling of HCN4. We generated expression constructs encoding full-length chick HCN4, the C-terminus of chick HCN4, the C-terminus of chick HCN1, the C-terminus of chick HCN2, and the C-terminus of mouse HCN4 and co-expressed these with a nuclear eGFP reporter in ventricular myocardium. The monoclonal antibody clone 2B2 successfully labeled both full-length chick HCN4 and the C-terminus of chick HCN4 when overexpressed, but did not label HCN1, HCN2 or mouse HCN4 (Fig S3B and C). We confirmed that this antibody detected endogenous HCN4 by staining cryosections of E6 cardiac tissue. The 2B2 clone successfully labeled the cell membrane of cardiomyocytes within the SAN but not in the atria. Importantly, the HCN4 2B2 staining pattern overlapped with the RNAscope-based detection of *Hcn4* transcript (Fig S3D and E).

### RNA sequencing

#### *RNA isolation*
HH 30 SAN and atrial free wall explants were mechanically isolated and pooled into three biological replicates as described in Fig S1A. Pooled samples were placed in a 1.5 ml Eppendorf tube containing sterile HBSS. The tubes were spun at 3,000 rpms for 3 min to loosely pellet tissue, HBSS was then aspirated away. 500 $\mu$l of TRIzol Reagent (#15596026; Invitrogen) was added samples were briefly

vortexed. Then, 200 μl of cold chloroform was added to the sample. The sample was inverted 10 times and allowed to incubate at room temperature for 10 min. Next, the sample was centrifuged at 12,000 rcf for 15 min at 4°C in an Eppendorf 5424 R bench top centrifuge (Rotor # FA-45-24-11). The aqueous phase was removed and added to a new 1.5 ml Eppendorf tube. Then, an equal amount of cold 70% Ethanol/DEPC-H2O was added. A Monarch Total RNA Miniprep Kit (#T2010S; New England BioLabs) was used to purify the RNA following the manufacturer's protocol. The samples were then transferred the UNC CGIBD Advanced Analytics Core for cDNA generation. Sequencing was completed using the Illumina NextSeq High Output Kitv2.5 kit and ran on the Illumina NextSeq 500 sequencer.

### Quality control, alignment, data analysis
The quality control and alignment were done in the Linux environment on the UNC Longleaf serve based on the work done by Love et al (2015). For quality control, FASTA files were run through FASTQC. The ends of the FASTA files were run through Trimmomatic using the recommended settings. The FASTA files were aligned to the *Gallus gallus* GRCg6a genome downloaded from the NCBI ensemble genome. SAM files were converted into a BAM File using the samtools package. The gene count matrix was created using the FeatureCounts package. Finally, the files were exported and downloaded from the Linux environment where the DSEQ2 analysis was completed in R. The differential expression was completed using the DSEQ2 package as previously described (Love et al, 2015).

### RNAscope (fluorescent in situ hybridization)

RNAscope was performed as described previously following the manufacturer's protocols (Thomas et al, 2021). For cells on hydrogels, CPCs were fixed with 10% NBF (HT5011; MilliporeSigma) followed by 3 × 10-min washes with DEPC-treated PBS at 4°C. Cultures were then dehydrated and rehydrated through a series of 10-min ethanol washes at 4°C. Afterward, samples were treated with RNAscope Hydrogen Peroxide Reagent (322330; ACD) for 10 min, rinsed in DEPC-water, and treated with RNAscope Protease III (1:15 in DEPC-PBS) for 10 min followed by DEPC-PBS washes and hybridization (322330; ACD). RNAscope was performed using Multiplex Fluorescent Reagent Kit v2 (323100; ACD). Probes were hybridized to cell culture samples for 2 h at 40°C. Both HCN4 (569141-C1, 1:1; ACD) and NCX1 (1005901-C3, 1:50; ACD) probes were designed by ACD against available NCBI chicken mRNA sequences. All RNAscope assays were performed with a positive control gene UBC (453961-C2, 1:50; ACD). Fluorescent dyes: TSA Plus Fluorescein, (NEL74001KT; Akoya Biosciences), TSA Plus Cyanine 3, (NEL744001KT; Akoya Biosciences), and TSA plus Cyanine 5, (NEL745001KT; Akoya Biosciences) were used at a 1:1,000 dilution in TSA buffer following the ACD RNAscope recommendation.

### Confocal microscopy

Standard resolution imaging was conducted using a Zeiss LSM800 upright confocal laser scanning microscope with 3 GaAsP confocal detectors. The following laser lines were used: 405, 488, 561, and 633 nm. The emission path on this system uses variable dichroic mirrors. The following objectives: PLN APO 25×/0.8 oil, PLN APO 63×/

1.4 oil, and N-APO 63×/0.9 water were used for the acquisitions. Acquisition was conducted using ZEN Blue Microscopy Software (Zeiss).

Super-resolution imaging was conducted using a Zeiss LSM 880 confocal laser scanning microscope with AiryScan. This system has a 34 channel GaAsP detector using laser lines: 405, 488, 561, and 633 nm. A PLN APO 63×/1.4 oil was used for acquisitions. ZEN Blue Microscopy Software (Zeiss) was used for data acquisition and super resolution processing.

### Transmission electron microscopy (TEM)

Samples were prepared for TEM as described previously (Thomas et al, 2021). Briefly, E6 embryos were fixed overnight with 4% PFA at 4C. The embryos were then rinsed and washed with 1X PBS before being embedded in 3% low melt agarose (Apex Chemicals and Reagents, 20–104) in 1X PBS. Vibratome sections were cut to 200 μm thick sections. These sections were then fixed at 4°C in 4% PFA, 1% glutaraldehyde, and 0.1 M sodium phosphate for 1–3 d. Sections were then transferred to 1% osmium tetroxide, 1.25% potassium ferrocyanide, 0.1 M sodium phosphate buffer. Sections were then dehydrated through an ethanol wash series (30%, 50%, 75%, 100%, 100%) and propylene oxide and embedded in PolyBed 812 epoxy resin (08792-1; Polysciences). Semi-thin 1 μm sections were cut and stained with 1% toluidine blue to determine the preservation and outline regions of interest by light microscopy. Samples were then cut to 70–80 nm thickness thin sections and mounted on 200 mesh copper grids and stained with 4% aqueous uranyl acetate for 12 min followed by lead citrate for 8 min. Sections were imaged on a Thermo Fisher Scientific (FEI) Tecnai 12 G2 transmission electron microscope operated at 80 kV using a 1 k × 1 k CCD camera (Model 794) mounted on a JOEL JEM-1230 transmission electron microscope.

### Image quantification

### Quantification of surface area, volume, and intensity
Imaging data was deconvolved using Autoquant (Media Cybernetics) and quantified using Imaris 5D image analysis software (Bitplane). For in vivo analysis, SAN tissue was transfected as described previously (Goudy et al, 2019; Thomas et al, 2021) with a DNA plasmid encoding a membrane localize RFP and Nuclear-targeted eGFP. Each cell was digitally isolated using the Imaris surface and mask selection feature. The RFP membrane reporter was then used to render the surface structure of the cell and surface area and volume data were exported. For CPCs in culture, the same workflow was used to create a 3D surface area and volumes for MF20, HCN4, and NCX1 staining. Staining intensity was calculated using ImageJ (NIH, V2.0.0). MF20 signal was used to create a region of interest outlining each cell and average signal intensity was calculated for HCN4 or NCX1 within that defined region of interest.

### Quantification of RNAscope data
RNAscope images were processed using ImageJ (V2.0.0; NIH). Briefly, max intensity z-projections were generated and a fixed intensity threshold applied to all cells. A region of interest was defined

**Table 1. Recipes for Polyacrylamide hydrogel preperation.**

| Stiffness (Pa) | Percent acrylamide | Percent bis-acrylamide | 40% acrylamide volume | 2% bis volume | 10X DPBS | Water | TEMED | 1% APS | Total volume |
|---|---|---|---|---|---|---|---|---|---|
| 200 | 3 | 0.060 | 150 | 60 | 200 | 1388 | 2 | 200 | 2,000 |
| 2,700 | 8 | 0.035 | 375 | 35 | 200 | 1188 | 2 | 200 | 2,000 |
| 22,000 | 8 | 0.250 | 375 | 250 | 200 | 973 | 2 | 200 | 2,000 |
| 50,000 | 10 | 0.500 | 500 | 500 | 200 | 598 | 2 | 200 | 2,000 |

outlining each CPC in and the analyze particles function was used to quantify the number of puncta per CPC.

## Mechanical probing and nanoindentation

The SAN and free wall of the right atria were removed from E6 hearts in 1x HBSS (14175095; Gibco) containing 25 mM 2,3-Butanedione 2-monoxime (BDM) (A14339.22; Thermo fisher Scientific) and 100 U/ml heparin (H0878; Sigma-Aldrich). To immobilize explants during mechanical probing, the tissue was adhered to a glass cover slip using 1% low melting temperature agarose. For macroscopic probing, samples were placed under an Orca Flash 4.0 CMOS camera (Hamamatsu) mounted on a Leica M165 stereo microscope. Tissues were probed with pulled thin-walled glass pipettes (TW100F-4; World Precision Instruments) with tip diameters between 20–30 $\mu$m. Glass pipettes were mounted on an Injectman four micromanipulator (Eppendorf). Pipette tips were placed in contact with the tissue and then a preprogramed indentation (150 $\mu$m travel, 200 $\mu$m/s) was performed. Tissue deformation was live-imaged at 100 frames per second.

Young's modulus was determined using a Piuma Nanoindenter (OpticsLife). Explants were prepared as described above and indentation was performed using an 8 $\mu$m diameter spherical force sensor with a stiffness of 0.35 N/m. Load/indentation curves were generated by indenting over a 3 × 3 matrix spanning 80 × 80 $\mu$m. The measurement protocol followed included a loading phase of 2,000 nm (after contact with the sample surface) and 1 s holding phase, and an unloading phase returning the force probe to ~8,000 nm above the tissue surface. Curve fitting and Young's Modulus calculations were performed in Dataviewer v2.4 (OpticsLife).

## CPC dissociation and culture

HH 30 SANs were dissected out of the heart and pooled in 1X HBSS (14175095; Gibco), supplemented with 15 mM HEPES (15630106; Gibco). SANs were dissociated for 30 min at 37°C in 17% Trypsin–EDTA (25200056; Gibco) in 1X HBSS supplemented with collagenase/dispase (1026963800; MilliporeSigma) at a final concentration of 8.5 $\mu$g/ml. Cells were then pelleted by centrifugation at 1,000 rcf for 5 min and washed three times in DMEM/12 (11330032; Gibco) supplemented with 15% FBS (97068-085; Avantor Seradigm), 1% penicillin–streptomycin (15140122; Gibco), and 1% amphotericin B (15290026; Gibco). Cells were then preplated (to deplete non-myocytes from the final cultures) for 45 min on glass coverslips coated with fibronectin (F2006; MilliporeSigma) at a concentration of 1 $\mu$g/cm$^2$. The supernatant was then removed and transferred to a fresh culture dish and CPCs were plated at a density of ~10$^6$ cells/cm$^2$. CPCs were maintained in culture for 48 h in DMEM/12

(11330032; Gibco) supplemented with 15% FBS (97068-085; Avantor Seradigm), 1% penicillin–streptomycin (15140122; Gibco), and 1% amphotericin B (15290026; Gibco).

## Hydrogel preparation

PA hydrogels were generated as previously described (Tse & Engler, 2010; Chopra et al, 2011; Thomas et al, 2021). Glass bottom 35 mm tissue culture dishes (D35-14-0-U; Matsunami Glass) were plasma-activated (PDC-001; Harrick Plasma) for 1 min and treated with 0.1% APTES (A3648; MilliporeSigma) for 30 min. Dishes were then washed three times with molecular biology grade water before fixing with 0.1% glutaraldehyde for 1 h (G5882; MilliporeSigma). Post fixation, the dishes were washed three times with DI water. Hydrogels of 200 Pa, 2.7, 22, and 50 kPa stiffnesses were generated using 10% acrylamide at 3%, 7.5%, 7.5%, 10% (A4058; MilliporeSigma) and bis-acrylamide at 0.06%, 0.035%, 0.25%, 0.5%, respectively (M1533; MilliporeSigma)—see Table 1. Hydrogels were polymerized with final concentrations of 0.1% TEMED (T9281; MilliporeSigma) and 0.1% APS (A3678; MilliporeSigma). The solution was pipetted onto the pre-prepared glass bottom 35 mm tissue culture dishes and covered with a quartz coverslip (CGQ060001; Chemglass Life Sciences) coated with Sigmacote (SL2; MilliporeSigma). Gels were polymerized for 10 min and then cross-linked in an UVO chamber for 4 min (Model 24; Jetlight Company, Inc.). Gels were submerged in water to remove the coverslip and the hydrogel surface substrate was activated with a solution of 17.5 mg/ml EDC (24500; Thermo Fisher Scientific) and 10 mg/ml NHS (22980; Thermo Fisher Scientific) in molecular grade water for 15 min. Gels were then rinsed with water and coated with fibronectin (F2006; MilliporeSigma) at a concentration of 100 $\mu$g/ml in 1X DPBS (14190144; Gibco) for 2–3 h at 37°C. The gels were then probed with a Piuma Nanoindenter (OpticsLife) using the same approach outlined above for tissue stiffness evaluation to determine their stiffness. After mechanical measurements, the gels were washed three times in 1X DPBS (14190144; Gibco) and sterilized by UV light.

## Calcium imaging

CPC cultures were loaded with 5 $\mu$M Calbryte-520 (20650; AAT Bioquest) in 1X HBSS (14175095; Gibco) supplemented with 15 mM HEPES (15630106; Gibco) for 30 min at 37°C with 5% $CO_2$. Cells were then transferred to a temperature-controlled stage insert (642415, 640110; Harvard Apparatus) at (34 ± 1°C) and allowed to recover for 1–5 min. Cultures were imaged with a Zeiss Axiovert 200 inverted fluorescence phase contrast microscope using an Acroplan 40×/0.8 W water-dipping objective, an X-CITE 120 LED Boost light source (Excelitas Technologies), and a Hamamatsu Orca-Flash4.0 V2 camera. Videos were recorded at 100 frames per second using

HCImage software (Hamamatsu). Imaging CPC calcium transients in vivo were conducting by co-transfecting embryonic chick hearts with DNA constructs encoding the genetic calcium reporter, gCAMP6f, and a membrane-targeted RFP as described previously (Goudy et al, 2019). Briefly, lipofectamine-encapsulated DNA plasmids were microinjected into the pericardial cavity of HH 18 embryonic chick hearts directly adjacent to the forming SAN. Embryos were then re-incubated to HH 30 and SANs were mechanically isolated and transferred to a heated imaging chamber mounted on a Nikon Eclipse Ti2 widefield automated microscope with an S Plan Flour LWD 20× objective (NA 0.7). Time series were recorded using a pc0.edge sCMOS camera (100 frames a sec at 2,048 × 2,048 resolution). All calcium transient analyser were performed using ImageJ.

### Voltage imaging/optical mapping

HH 30 Tissue explants were isolated in 1X HBSS (14175095; Gibco) supplemented with 15 mM HEPES (15630106; Gibco) at 37°C. The tissues were allowed to recover for 30 min at 37°C in a tissue culture incubator supplemented with 5% $CO_2$. Explants were loaded for 10 min with 10 $\mu$M Di-4-ANEPPS (D1199; Invitrogen) and 10 $\mu$M (−) Blebbistatin (B0560; MilliporeSigma) at 37°C with 5% $CO_2$. Explants were then placed in a temperature-controlled imaging chamber maintained at 36 (± 1°C) containing 1 X HBSS (14175095; Gibco) and perfused with HBSS saturated with 95% $O_2$ and 5% $CO_2$. Imaging was performed on a vertical THT fluorescent widefield macroscope (SciMedia) with an LED LEX2 light source (SciMedia) and a green excitation filter cube (SciMedia, DLFLSP2R Beam Splitter). Action potentials were recorded at 2,000 frames per second using a 14-bit, 100 × 100-pixel, CMOS Camera (MiCAM05 Ultima; SciMedia). The two air objectives used were a PLAN APO 5.0x/0.50 LWD (10447243; Leica) and a PLAN APO 2.0X (10450030; Leica). BV-Ana software (SciMedia) was used to calculate velocity, cycle length, and $APD_{70}$.

### Statistical analysis

For all studies, biological replicates and/or cell numbers are reported in the relevant figure legend. For all analyses, Mean ± SD is reported. Data distribution was determined using Prism (Graphpad). All variables were considered as independent and statistical significance was calculated using a two-sample unpaired $t$ test (*<0.05, **<0.01, ***<0.005, ****<.001).

## Data Availability

The RNA sequencing data from this publication have been deposited in the Gene Expression Omnibus (GEO) database under the accession number GSE227027.

## Supplementary Information

## Acknowledgements

We thank M Joseph Costello for his technical expertise related to ultrastructural analysis. Microscopy services were provided by the Hooker Imaging Core Facility at the University of North Carolina (supported NIH-NINDS P30 NS045892) and the Neuroscience Microscopy Core at the University of North Carolina (supported NIH-NINDS P30 NS045892 and NIH-NICHD U54 HD079124). RNA sequencing support was provided by the UNC CGIBD Advanced Analytics Core. In addition, we thank the Chapel Hill Analytical and Nanofabrication Laboratory core facility (CHANL). CHANL is a member of the North Carolina Research Triangle Nanotechnology Network, RTNN, which is supported by the National Science Foundation, Grant ECCS-2025064, as part of the National Nanotechnology Coordinated Infrastructure, NNCI. This work was supported by National Institutes of Health grants R01HL146626 (NHLBI) to M Bressan and K08HL152308 to R Wirka.

### Author Contributions

T Henley: data curation, formal analysis, investigation, methodology, and writing—original draft, review, and editing.
J Goudy: data curation, formal analysis, and writing—original draft, review, and editing.
M Easterling: data curation, formal analysis, methodology, and writing—review and editing.
C Donley: data curation and formal analysis.
R Wirka: data curation, formal analysis, and investigation.
M Bressan: conceptualization, data curation, formal analysis, funding acquisition, investigation, methodology, project administration, and writing—original draft, review, and editing.

### Conflict of Interest Statement

The authors declare that they have no conflict of interest.

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
