## [Reviewer comments · Life Science Alliance]

Local Tissue Mechanics Control Cardiac Pacemaker Cell Embryonic Patterning

Trevor Henley, Julie Goudy, Marietta Easterling, Carrie Donley, Robert Wirka and Michael Bressan

DOI: 10.26508/lsa.202201799

Corresponding author(s): Dr. Michael Bressan (University of North Carolina at Chapel Hill)

Review timeline:

Submission Date:	2022-11-03
Editorial Decision:	2022-12-08
Revision Received:	2023-02-10
Editorial Decision:	2023-03-06
Revision Received:	2023-03-13
Accepted:	2023-03-15

Scientific Editor: Eric Sawey

Transaction Report:

No Peer Review Process File is available with this article, as the authors have chosen not to make the review process public in this case.

Re: Life Science Alliance manuscript #LSA-2022-01799-T

Dr. Michael Bressan
University of North Carolina at Chapel Hill
Cell Biology and Physiology
111 MASON FARM RD
MBRB Rm 6341C
Chapel Hill, NC 27599

Dear Dr. Bressan,

Thank you for submitting your manuscript entitled "Local Tissue Mechanics Control Cardiac Pacemaker Cell Embryonic Patterning." to Life Science Alliance. The manuscript was assessed by expert reviewers, whose comments are appended to this letter. We invite you to submit a revised manuscript addressing the Reviewer comments.

Thank you for this interesting contribution to Life Science Alliance. We are looking forward to receiving your revised manuscript.

Sincerely,

Eric Sawey, PhD
Executive Editor
Life Science Alliance
<http://www.lsa-journal.org>

-- An editable version of the final text (.DOC or .DOCX) is needed for copyediting (no

PDFs).

B. MANUSCRIPT ORGANIZATION AND FORMATTING:

RE: Life Science Alliance Manuscript #LSA-2022-01799-TR

Dear Dr. Bressan,

Thank you for submitting your revised manuscript entitled "Local Tissue Mechanics Control Cardiac Pacemaker Cell Embryonic Patterning.". We would be happy to publish your paper in Life Science Alliance pending final revisions necessary to meet our formatting guidelines.

- please add ORCID ID for both corresponding authors-you should have both received instructions on how to do so
- please add the Twitter handle of your host institute/organization as well as your own or/and one of the authors in our system
- please add a separate conflict of interest statement to the main manuscript text
- please use the [10 author names, et al.] format in your references (i.e. limit the author names to the first 10)
- please add figure callouts for Figure 3F,G; Figure 4D; and Figure S4

A. FINAL FILES:

-- Summary blurb (enter in submission system): A short text summarizing in a single sentence the study (max. 200 characters including spaces). This text is used in conjunction with the titles of papers, hence should be informative and complementary to the title. It should describe the context and significance of the findings for a general readership; it should be written in the present tense and refer to the work in the third

person. Author names should not be mentioned.

B. MANUSCRIPT ORGANIZATION AND FORMATTING:

Sincerely,

3rd Editorial Decision

15 March 2023

RE: Life Science Alliance Manuscript #LSA-2022-01799-TRR

Dr. Michael Bressan
University of North Carolina at Chapel Hill
Cell Biology and Physiology, McAllister Heart Institute
111 MASON FARM RD
MBRB Rm 6335
Chapel Hill, NC 27599

Dear Dr. Bressan,

Thank you for submitting your Research Article entitled "Local Tissue Mechanics Control Cardiac Pacemaker Cell Embryonic Patterning.". It is a pleasure to let you know that your manuscript is now accepted for publication in Life Science Alliance. Congratulations on this interesting work.

DISTRIBUTION OF MATERIALS:

Again, congratulations on a very nice paper. I hope you found the review process to be constructive and are pleased with how the manuscript was handled editorially. We look forward to future exciting submissions from your lab.

Sincerely,
